# Analysis of Soft Skills and Job Level with Data Science: A Case for Graduates of a Private University

Sofía Ramos-Pulido [1,*] , Neil Hernández-Gress [1] and Gabriela Torres-Delgado [2]

1 School of Engineering and Science, Tecnologico de Monterrey, Monterrey 64849, Mexico
2 School of Humanities and Education, Tecnologico de Monterrey, Monterrey 64849, Mexico
* Correspondence: a00834077@tec.mx

**Abstract:** This study shows the significant features predicting graduates' job levels, particularly high-level positions. Moreover, it shows that data science methodologies can accurately predict graduate outcomes. The dataset used to analyze graduate outcomes was derived from a private educational institution survey. The original dataset contains information on 17,898 graduates and approximately 148 features. Three machine learning algorithms, namely, decision trees, random forest, and gradient boosting, were used for data analysis. These three machine learning models were compared with ordinal regression. The results indicate that gradient boosting is the best predictive model, which is 6% higher than the ordinal regression accuracy. The SHapley Additive exPlanations (SHAP), a novel methodology to extract the significant features of different machine learning algorithms, was then used to extract the most important features of the gradient boosting model. Current salary is the most important feature in predicting job levels. Interestingly, graduates who realized the importance of communication skills and teamwork to be good leaders also had higher job positions. Finally, general relevant features to predict job levels include the number of people directly in charge, company size, seniority, and satisfaction with income.

**Keywords:** graduates; soft skills; job level; gradient boosting; random forest; decision trees; regression; predict; important features

## 1. Introduction

Many university students have high expectations for life and their career after school. However, the job market today is more competitive than ever, and educational institutions now require more compromise. Currently, many universities worldwide are assessing their impact through their graduates [1–3]; for instance, institutions gather data on these graduates' college satisfaction, employment, entrepreneurship, innovation, volunteerism and mentorship.

According to the existing literature, job level is considered an indicator of career success [4] as well as salary. Salary and job level are related, as will be shown in the Results section. Refs. [5,6] found that job level determines the differences in earnings or variations in pay. Ref. [7] reported that salary is a sensitive question in a survey. We believe that asking the job level in a survey implied less bias in the answer than asking about the salary. Therefore, in this work, we will use job level as a target variable to measure career success. This study employed a survey conducted in 2018 by a private university to measure their graduates' social and economic impact. This study's primary purpose is to determine whether there is a relationship between job levels and soft skills, such as communication, teamwork, planning, and innovation. Furthermore, we aim to show that data science methodologies accurately predict job levels.

According to an American worldwide employment website for job listings [8], typical job levels may be defined as executive or senior management, middle management, first-level management, intermediate or experienced, and entry level. Executive or senior management includes the chief officers, president, vice president, senior executive,

and executive. Middle management includes the senior director, director, associate director, regional manager, and adviser. First-level management includes the senior manager, manager, supervisor, project manager, team leader, and office manager. Intermediate or experienced employees include the coordinator, analyst, and specialist. Entry–level employees include the staff member, representative, and associate.

The study first contributes by adding to the existing literature on job levels. Second, it contributes to an application in the field of data science by exploring algorithms such as decision trees, random forest, and gradient boosting and by comparing them with traditional regression models such as ordinal regression. Third, it contributes to the application of explainable artificial intelligence by identifying important features with SHapley Additive exPlanations (SHAP) values. Finally, it contributes to students and educational institutions in Mexico by providing findings relevant to graduates. It is important to mention that to our knowledge, only one study by [9] has analyzed graduates' outcomes, covering research on the alumni income of a private Mexican institution.

This paper is organized into several sections. Related studies are shown in the next section. Then, we summarize the methodology. We show the adjusted models and the most important features to predict three job levels in the Results section. Finally, we present the conclusion.

## 2. Review of Related Studies

Most of the studies on job level aim to understand the relationship between job level and job satisfaction. Under some conditions, some studies have found a relationship between job level and job satisfaction [10–15]. Ref. [12] found that job satisfaction increased as job levels increased. This positive relationship was also observed for higher work levels by [10]. Ref. [15] showed that exhaustion, cynicism, and professional inefficacy mediate the relationship between job level and job satisfaction. Furthermore, ref. [16] mentioned that people at higher job levels are more satisfied with their work and pay. In this study, we present features that measure satisfaction with career and income, but we did not find a high correlation with job level.

The general findings related to job level include associations with family conflicts, stress, and age. For work–family conflict, ref. [17] found that higher-level employees experience more work interference with family than those at lower levels of the organization. Likewise, higher-level workers experience more family interference with work. These two associations were explained by their extensive job demands and work hours. Similarly, ref. [18] discovered that those in professional positions with high job demands were most likely to experience work–family stress. Ref. [19] found that the impact of coworker relations on intent to stay was stronger for employees in high-status positions. Finally, ref. [20] showed that age is negatively correlated with job level.

Recently, becoming acquainted with the skills that people need for good job positions has had an increasing trend. However, this topic has been previously studied. Ref. [21] showed that higher-level managers emphasized self-actualization and autonomy needs more than lower-level managers. Ref. [22] found that job level was related to cognitive differentiation, self-monitoring, perspective-talking, and persuasive ability. Ref. [23] reported that supervisors received more positive relationships and upward openness in communication. More recently, ref. [24] showed that problem-solving and general mental ability partly contributed to the variance in job level.

Ref. [25] found that executives had a personality profile characterized by low levels of social inhibition, feelings of inadequacy, and eagerness to please. They possessed high levels of expressiveness, need for attention, self-confidence, higher scores on a scale measuring creativity, and unusual thinking. The study also indicated that organizational leaders are characterized by an interpersonal orientation dominated by assertive, self-assured, and sociable behavior and low levels of interpersonal insecurity, skepticism, and the need to please others.

We can find a lot of information about senior-level skills on the internet. The University of [26] has provided a list of skills for senior-level leadership that is recommended to be developed or improved. This list includes leadership skills, communication and presentation skills, management skills, strategic thinking and foresight, decision-making, emotional intelligence, employee development, and delegation. Ref. [27] considers four fundamental leadership skills, self-awareness, communication, influence, and learning agility. On LinkedIn, ref. [28] stated that managers ought to have a mix of leadership and managerial skills, learning agility, delegation, strategic considerations, influencing communication, thorough marketing knowledge, employee training, and development. As previously mentioned, there are numerous sources, but most of them frequently mention leadership, communication, delegation, and learning.

## 3. Materials and Methods

The principal material of this research is the database from a graduate survey of a Mexican private university. Data preparation or dataset was an essential process that included several steps. First, the original dataset was cleared of inconsistencies and outliers. Second, we worked with missing values. We deleted features with over 80% missing data that were unrelated to the target variable. We treated these records and then used mean or k-nearest neighbors to complete the actual missing data. Third, data binning was used to reduce the imbalance in some features. For example, graduates' birth states can include more than 20 states. These states were grouped into regions. Fourth, we coded: nominal features such as level of studies and dichotomous variables and created dummy features for nominal variables. Last, data standardization was performed with the normalization method.

Once the data was prepared, we used descriptive-correlational analysis and chi-squared tests to show relationships and associations between job level and other features. Then, three supervised learning models for classification were implemented to show that the data science models can accurately predict job levels. The data mining algorithms studied were decision trees, random forest, and gradient boosting. We select decision and ensemble trees because we need an explainable model to extract the most important features for every level of the target variable. With the best predictive model, the most important features were selected with SHAP (SHapley Additive exPlanations).

### 3.1. Sample

The sample comprised graduates from different campuses and schools of the Tecnológico de Monterrey university. The survey invitation was electronically sent to all graduates between 1953 and 2017 (269,482 records), with a 7% response rate. Tecnológico de Monterrey provided the dataset without personal identification. While the original dataset contained information on 17,896 graduates and 148 features (including the dummy variables), the final dataset contained information on 11,969 graduates and 121 features. Around 33% of the respondents did not respond to the target variable, job level. Therefore these records were eliminated.

More men (57%) responded to the survey than women (43%). The age group with the highest frequency was that between 30 and 39 years (34%), followed by 40–49 years (25%), less than 30 years (25%), between 50 and 60 years (12%), and over 60 years (4%). Regarding the school to which they belonged, 42% were from the engineering and science school, 37% were from the business school, and the rest 21% were from humanities and education, government and social science, architecture, and medicine. We note that the majority (69%) resided in the center and north of the country and a significant percentage (17%) resided abroad.

### 3.2. Instrument and Measures

The survey contained 50 questions, and most were dichotomous or multiple choice. The survey content was validated by an inter-judge agreement and research professors who had previously validated the item content and form.

#### 3.2.1. Input Features

The total number of input features is 121. These features include initial and current salary, number of people in direct charge, age, gender, current address, region of birth, parents' education, working hours, years of working abroad, evaluations of the importance of communication, teamwork, innovation, and planning to be a good leader, etc.

The number of people in charge was asked in the survey at intervals. Therefore, this feature was codified as follows: the low level corresponds to graduates without people directly in charge, level two corresponds to between 1 and 10 people in charge, and level three corresponds to more than 10 people directly in charge.

The current salary is grouped into levels corresponding to salary quantiles. The first level contains salaries lower than the 25% quantile, the second level corresponds to salaries between the 25% and 50% quantiles, the third level corresponds to salaries between the 50% and 75% quantiles, and the fourth level corresponds to salaries higher than the 75% quantile.

#### 3.2.2. Target Variable

There are several ways to classify job levels. Most of the research used three or four job levels. For example, ref. [29] defined top management as the president, vice president, and the overall government of the organization. Middle management: persons who operate divisions or departments. Lower management includes those persons putting into effect the procedures developed by middle management.

Taking four levels, ref. [10] considered workers and technicians in the low job level, first-line supervisors in the second job level, co-officer and the department head in the middle job level, and top management level in the last job level. Another reference is [5] using three levels. They considered the high job level containing executives, managers, and professionals. The medium job level included clerks, technical workers, and assistant professionals, and the low job level considered first-line service workers, machine operators, manual laborers, and farm and fishery workers. More recently, ref. [25] considered three groups for job level, entry-level supervisors, middle managers, and executives.

We coded job levels with a similar classification used in previous studies, especially for high and middle job levels. Job level was created using the ten categories of the question on the current job position. Level one comprises the department head, teacher, salesperson, and independent professional; level two comprises the consultant, regional manager, and director; and level three includes the owner of a business, CEO/general manager, and VP/deputy manager.

### 3.3. Data Analysis

The job level, with three predefined categories, was studied. Data science methods were the supervised methods for classification. Machine learning algorithms included decision trees, random forest, and gradient boosting, whereas ordinal regression was used as a baseline method to compare the results of machine learning algorithms.

One purpose of the investigation was to show that data science models can provide accurate predictions of job levels. Nevertheless, at the same time, we needed it to easily extract the important features for every level of the target variable. Decision trees, random forests, and gradient boosting were selected because they are recognized for good predictive and interpretable models.

Decision trees are a supervised data mining algorithm that classifies the classes of a target variable using features associated with the target variable. The features form a hierarchy encoded as a tree, where each feature is contained in a node, and each internal node points to a child node of the feature's classes [30]. When building a classification

tree, either the Gini index or the cross-entropy are typically used to evaluate the quality of a particular split [31]. We used DecisionTreeClassifier of [32] and Grid Search CV to evaluate criterion's Gini and entropy, and max depth. For max depth, the values 1 to 8 were evaluated.

The second data science algorithm used was random forests, random forest builds many decision trees on bootstrapped training samples [31], and the predictions of the resulting ensemble of decision trees are combined by taking the most common forecast [30]. Finally, gradient boosting, which is similar to the random forest algorithm, combines multiple simple decision trees but does not use bootstrap sampling; trees are grown using information from the previously grown tree [31]. Again, functions of [32] were used to fit the models and to evaluate important parameters like n of estimators and max depth. For max depth, the values 1 to 8 were evaluated. For the number of estimators, the list [20,40,60,80,100,120,140,160] was evaluated.

To evaluate the performance of the model, random cross-validation (CV) was used by creating five random splits of the complete data set into a training set (70%) and a testing set (30%). For each of these splits, the machine learning models were fitted using the training set, and their predictive performance was evaluated using the testing set. An average performance of the model was obtained by averaging over the five splits used.

In the training process, the best values of the hyperparameters were selected by using 5-Fold cross-validation (CV) for each model and each split in the random CV. Where the training set (70% of the data) was partitioned into five almost equal subsamples, and 4 out of these five subsamples were used to fit the model on each value of the selected grid for the hyperparameter, and the resulting models were evaluated in the left out subsample. This last process was repeated five times until each subsample (1/5) was left out, and then for each element of the grid, an average value was computed over the five repetitions, which was taken as the validation metric value for each value in the grid. Then, the "optimal" values of hyperparameters were chosen as the values in the grid with the highest value in the validation metric. Finally, these optimal values were used to fit the model again with the complete training set data, and then the testing set was predicted.

Several metrics were assessed to evaluate the performance of the models. These metrics include accuracy, recall, precision, F1, and AUC, and their formulas are described below.

Accuracy measures the number of times the classifier made the correct prediction.

$$accuracy = \frac{(TP + TN)}{(TP + TN + FP + FN)} \tag{1}$$

where:

- $TP$ is true positive
- $TN$ is true negative
- $FP$ is false positive
- $FN$ is false negative.

Precision measures the ratio of correct positive predictions to the total number of positive predictions. Recall is the ratio of accurate positive predictions to the dataset's overall number of positive observations.

$$recall = \frac{TP}{(TP + FN)} \tag{2}$$

$$presicion = \frac{TP}{(TP + FP)} \tag{3}$$

The *F*1 score combines the precision and recall values and tends toward the smaller value of the two elements.

$$F1 = \frac{2(precision \times recall)}{(precision + recall)} \tag{4}$$

AUC is a metric that measures the area under an ROC (receiver operating characteristic) curve. The ROC graph represents the relative tradeoffs between *TP* and *FP*, [33]. According to [33], the AUC score is equivalent to the probability that the classifier will rank a randomly chosen positive instance higher than a randomly chosen negative instance. It was also mentioned that an AUC of less than 0.5 depicts no realistic classifier.

After the adjustment and validation of the models, the metrics were compared to select the best predictive model. The model with the best metrics was selected. This model is fixed here as the best model to predict the target variable, job level.

To extract the most significant features of the best predictive model, SHapley Additive exPlanations (SHAP) was used. SHAP provides important features from different machine learning algorithms, relationships among feature values, and interpretations of multiclass classifiers [34]. According to [35], the SHAP method unifies six existing methods and three desirable properties (local accuracy, missingness, and consistency).

In this work, two types of SHAP graphs are used, the feature importance plot, Figure 1, and the SHAP summary plot, Figure 2. According to Ref. [36], global feature attribution is represented by four methods: the mean magnitude of the SHAP values, gain, split count, and feature permutation. The TreeExplainer of [37] was used to estimate SHAP values.

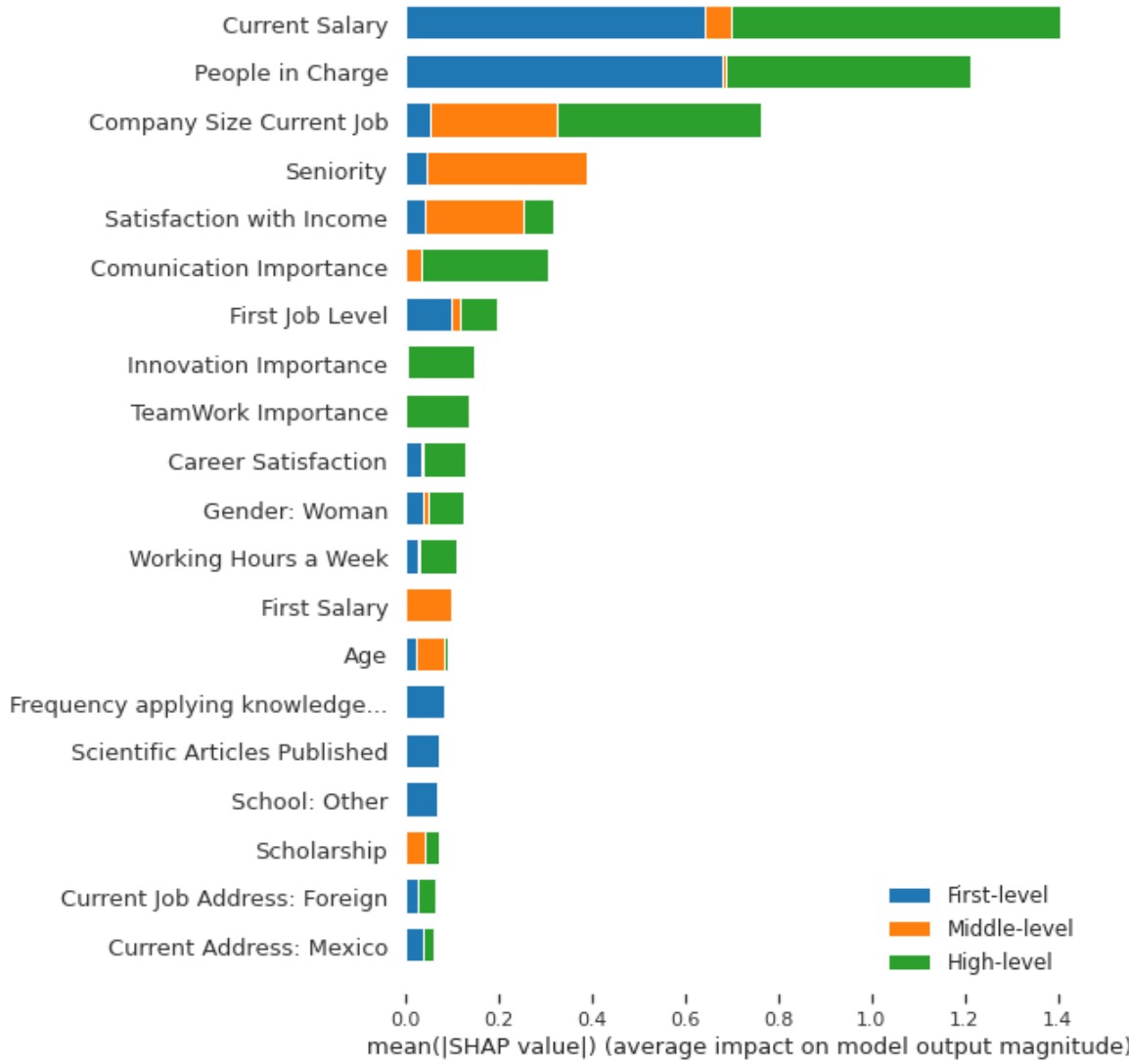

**Figure 1.** Feature importance for job level.

In the SHAP summary plot, the features are sorted by their global impact on the y-axis, and on the x-axis, the dots represent the SHAP values. The dots are colored accordingly to the feature, with low values (blue) to high values (red). For example, in Figure 2 that displays important features to predict the first job level, the variable people in charge possess the greatest SHAP value. The color red represents graduates with more people in charge, and the blue color represents graduates with few people in charge. The interpretation could be that graduates with fewer people in charge increase SHAP values or better predict a first job level.

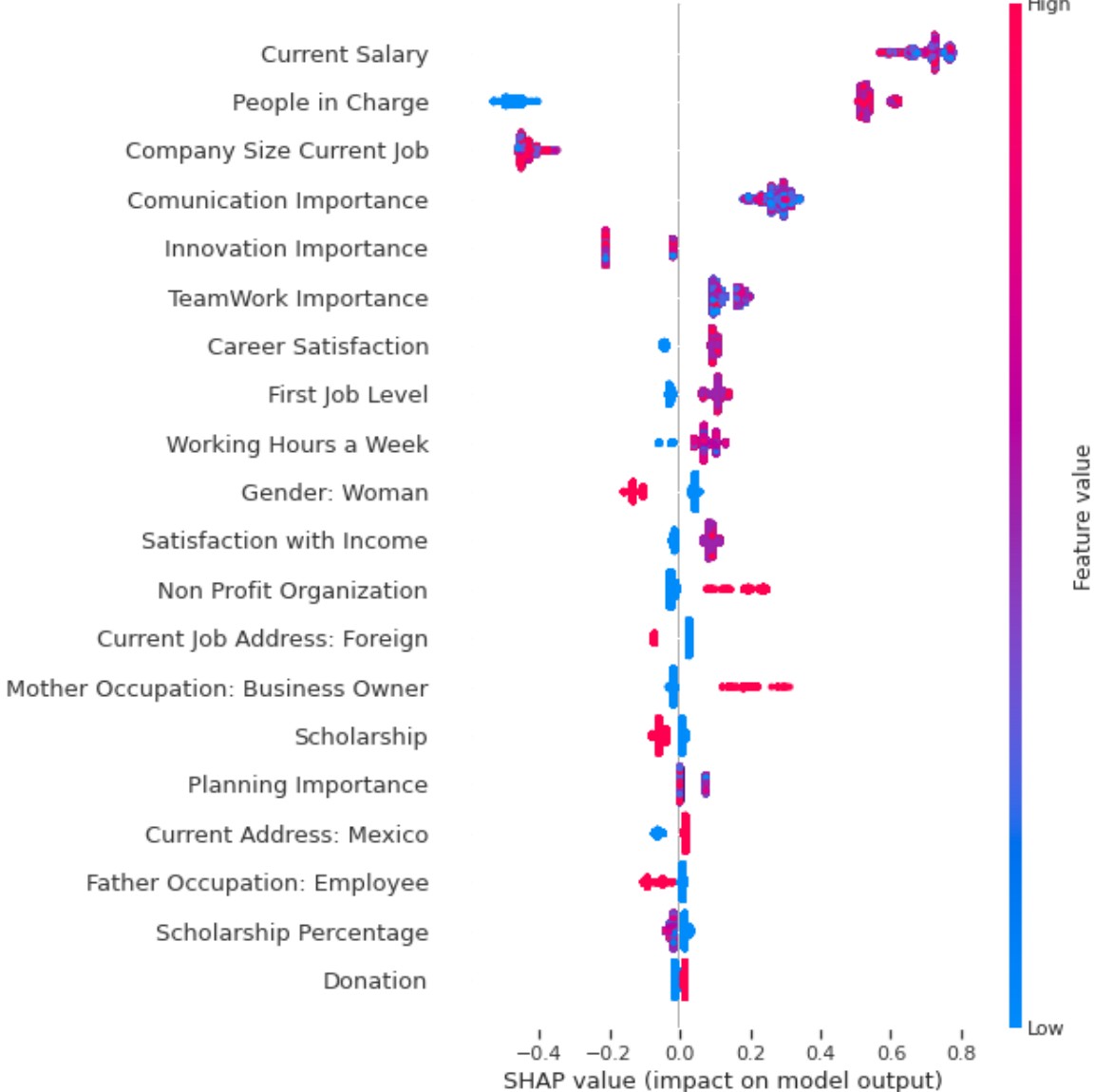

**Figure 2.** First job level.

## 4. Results

### 4.1. Correlational Analysis and Chi-Squared Test

Only two variables are moderately correlated with actual job levels. Job level is correlated with the number of people in charge at 0.41. Moreover, it is moderately correlated with the current salary at 0.38. As presented in Table 1, it is possible to see the correlations and the results of the chi-squared tests between job level and other features. Only correlations of at least 0.15 and statistically significant associations are shown.

**Table 1.** Correlations and Chi-squared tests.

| Metric/Model | Correlation | *p*-Value | $\chi^2$ | *p*-Value |
|---|---|---|---|---|
| People in charge | 0.41 | 0.00 *** | 2161.2 | 0.00 *** |
| Current salary | 0.38 | 0.00 *** | 2005.8 | 0.00 *** |
| Company size | −0.29 | $9.2 \times 10^{-229}$ *** | 1991.1 | 0.00 *** |
| First salary | 0.17 | $9.0 \times 10^{-77}$ *** | 402.6 | $7.5 \times 10^{-84}$ *** |
| Career satisfaction | 0.17 | $1.2 \times 10^{-74}$ *** | 403.8 | $4.2 \times 10^{-86}$ *** |
| Income satisfaction | 0.17 | $3.7 \times 10^{-77}$ *** | 394.8 | $3.8 \times 10^{-84}$ *** |
| Working hours a week | 0.15 | $1.2 \times 10^{-64}$ *** | 833.8 | $4.5 \times 10^{-82}$ *** |

Note: *** $p < 0.01$.

The association between job level and people in direct charge and that between job level and current salary are directly proportional. We can see these tendencies in Table 2. Note that current salary is grouped into levels corresponding to the salary quantiles. When the salary is low, the job level is low, and as the salary increases, the chances of having a high job level increase. Similarly, when the level of people in charge increases, the job level increases, and when the level of people in charge decreases, the job level decreases. More people in charge imply more possibilities to possess a high job level.

**Table 2.** Job levels and correlated features.

| Job Levels | Current Salary | | | | People in Charge | | |
|---|---|---|---|---|---|---|---|
| | 1 | 2 | 3 | 4 | 1 | 2 | 3 |
| First | 65% | 39% | 26% | 12% | 66% | 31% | 15% |
| Middle | 19% | 40% | 37% | 37% | 26% | 38% | 35% |
| High | 16% | 21% | 37% | 51% | 8% | 31% | 50% |

*4.2. Supervised Learning Models*

Five methods were implemented to extract the best predictive model and the most important features to predict job level. As mentioned, the Sklearn library functions fit the decision trees, random forest, gradient boosting, and logistic regression algorithms. Statsmodels were used for ordinal regression.

The max depth for decision trees, gradient boosting, and random forest selected were respectively 7, 2, and 4 for each method. Particularly in decision trees, to select which attribute to prove in each tree node, two attributes were tested: entropy and Gini. We select Gini because it provides the best results. Last, for the number of estimators, the selected values were 100 for gradient boosting and 140 for the random forest.

Table 3 displays the test set's accuracy, precision, recall, F1, and AUC results. The results are similar. It can be seen that gradient boosting is the best predictive model. Accuracy, precision, recall, and F1 are four points above logistic regression and six points more than ordinal regression. At the same time, AUC is five points above ordinal regression and 3 points more than logistic regression.

**Table 3.** Weighted average metrics.

| Metric/Model | DT | GB | RF | LR | OR |
|---|---|---|---|---|---|
| Accuracy | 0.61 | 0.67 | 0.62 | 0.63 | 0.61 |
| Precision | 0.61 | 0.67 | 0.62 | 0.63 | 0.61 |
| Recall | 0.62 | 0.67 | 0.62 | 0.63 | 0.61 |
| F1 | 0.61 | 0.67 | 0.62 | 0.63 | 0.61 |
| AUC | 0.78 | 0.83 | 0.80 | 0.80 | 0.78 |

DT: Desicion trees; GB: Gradient boosting; RF: Random forest; LR: Logistic regression; OR: Ordinal regression.

In the next section, in order of importance, we display the most important features in general and for every level of the target variable using gradient boosting.

## 5. Important Features with Gradient Boosting

The current salary, people in charge, company size of the current job, seniority, satisfaction with income, communication importance, first job level, innovation importance, teamwork importance, and career satisfaction are among the 10 most important features to predict the classes of the target variable, that is, current job level (Figure 1).

Figures 2 and 3 show the most important features to predict the low and high classes of job level. The people in charge, current salary, and first job level are the three most important features of the first job level; seniority, company size of the current job, and satisfaction with income are the three most important features of the middle job level (see Figure A1 in the Appendix A); and the current salary, people in charge, and company size of the current job are the three most important features of the high job level.

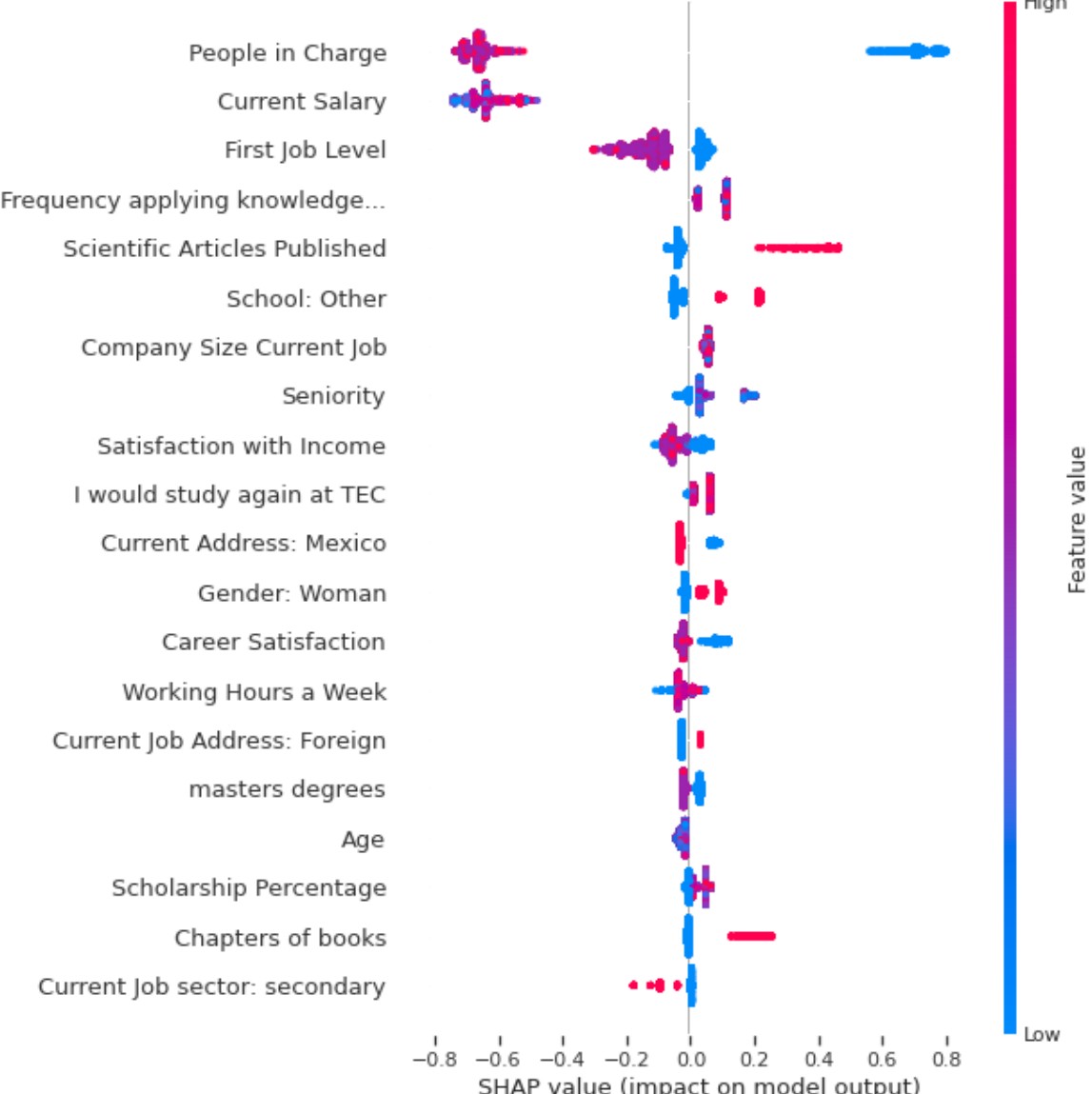

**Figure 3.** High job level.

Figures 2 and 3 show several positive effects or tendencies. Graduates with more people in charge very often have a high job level. In contrast, graduates who do not have

people in charge often have low job levels. Similarly, it is distinguished by the help of seniority to discriminate between job levels. People working for fewer years in companies are very often in the middle or low classes of the job level. Even satisfaction with income and career increases the job level. It can be seen that people with less satisfaction are often at a first job level, and people with more satisfaction are often at a high job level. On the other hand, it can be seen that when the first job level is at the top, it will remain at the top of the current job level. Moreover, if the first job level was at the lowest, the current job level could remain at the lowest.

As shown in Figures 2 and 3, the effect of the features, i.e., current salary and company size of the current job, needs to be clarified. Using the results of the correlational analysis shows that when the salary is the lowest, the job level is the lowest. Moreover, as the salary increases, the job level also increases. For the contribution of the organization's size in predicting job level, as presented in Table 4, the job level is usually intermediate or lower when graduates work in a big company. In contrast, a high job level is more common if the organization has fewer employees. This is probably because many of the analyzed university's graduates own businesses.

Some fascinating findings for the high job level are related to the evaluation communication and teamwork skills to be a good leader. Using Figures 2 and 3 and Table 4, people who better evaluate communication and teamwork are often at a high job level. Approximately 57% of those who evaluated communication as important and 43% of those that considered teamwork critical are graduates in high-level positions. There are no significant differences for the importance of innovation. Regarding age, the job level increases as age increases. Finally, the general results indicate that people working more hours a week are often at a high job level.

**Table 4.** Job level vs. important features.

| Features | | Job Level | | | Features | | Job Level | | |
|---|---|---|---|---|---|---|---|---|---|
| | | **First** | **Middle** | **High** | | | **First** | **Middle** | **High** |
| Salary | $[0, q_{25}]$ | 65% | 19% | 16% | Company size | $[1, 10]$ | 27% | 12% | 61% |
| | $[q_{25}, q_{50}]$ | 39% | 40% | 21% | | $[11, 50]$ | 21% | 26% | 53% |
| | $[q_{50}, q_{75}]$ | 26% | 37% | 27% | | $[51, 100]$ | 38% | 38% | 24% |
| | $[q_{75}, q_{100}]$ | 12% | 37% | 51% | | $\geq 100$ | 41% | 42% | 17% |
| Communication $^{imp}$ | 1 | 39% | 36% | 25% | Teamwork $^{imp}$ | 1 | 35% | 36% | 29% |
| | 2 | 38% | 36% | 26% | | 2 | 35% | 36% | 29% |
| | 3 | 37% | 36% | 27% | | 3 | 38% | 35% | 27% |
| | 4 | 31% | 29% | 40% | | 4 | 33% | 30% | 27% |
| | 5 | 28% | 30% | 42% | | 5 | 35% | 31% | 34% |
| | 6 | 15% | 28% | 57% | | 6 | 32% | 26% | 43% |
| Innovation $^{imp}$ | 1 | 37% | 26% | 37% | Age | 1 | 58% | 26% | 16% |
| | 2 | 33% | 31% | 36% | | 2 | 35% | 40% | 25% |
| | 3 | 37% | 34% | 29% | | 3 | 27% | 36% | 37% |
| | 4 | 34% | 34% | 32% | | 4 | 28% | 27% | 45% |
| | 5 | 36% | 37% | 27% | | 5 | 28% | 27% | 46% |
| | 6 | 34% | 38% | 28% | | | | | |

$^{imp}$ Evaluation to communication, innovation or teamwork importance.

## 6. Conclusions

First, in this study, we found that soft skills are essential features for predicting high-level positions. Graduates who realize the importance of communication and teamwork

skills to be good leaders are in high positions in their jobs. Additional features predicting job level include current salary and the number of people directly in charge, and both these features have a positive relationship with job level. For instance, when the salary is low, the job level is low, and a high compensation usually indicates a high job level.

For the second part of our work, it can be mentioned that machine learning algorithms predict job levels better than baseline ordinal and logistic regression. Gradient boosting was the best model, which was approximately 4–6 percentage points better than ordinal and logistic regression.

However, this study is limited to the number of features and the answers of the surveyed participants from a particular private university. Future research should include more features to investigate the relationships between job level and other soft skills, such as empathy, emotional intelligence, leadership, discipline, and honesty. It would be interesting to understand the similarities and differences among graduates of different universities.

To conclude, there are some actionable features that can improve the possibility of having a high-level position. Refs. [38–40] stated that communication or teamwork skills are trainable. As mentioned before, the last two skills are significant in predicting high-level positions. Therefore, institutions should develop ways to help students improve these and other essential soft skills.

**Author Contributions:** Conceptualization, S.R.-P., N.H.-G. and G.T.-D.; Methodology, S.R.-P. and N.H.-G.; Data curation, S.R.-P. and N.H.-G.; Formal analysis, S.R.-P.; Visualization, S.R.-P.; Writing—original draft, S.R.-P. and N.H.-G.; Writing—review & editing, S.R.-P., N.H.-G. and G.T.-D. All authors have read and agreed to the published version of the manuscript.

**Funding:** This research received no external funding.

**Institutional Review Board Statement:** Not applicable.

**Informed Consent Statement:** Not applicable.

**Data Availability Statement:** Not applicable.

**Acknowledgments:** The authors show appreciation to the Tecnologico de Monterrey for allowing them to use the database of graduates of the 75th anniversary.

**Conflicts of Interest:** The authors declare no conflict of interest.

## Abbreviations

The following abbreviations are used in this manuscript:

| | |
|---|---|
| DT | Decision trees |
| GB | Gradient boosting |
| RF | Random Forest |
| LR | Logistic Regression |
| OLR | Ordinal Logistic Regression |
| SHAP | SHapley Additive exPlanations |
| Acc | Accuracy |
| CV | Cross Validation |
| ROC | Receiver operating characteristics |

**Appendix A**

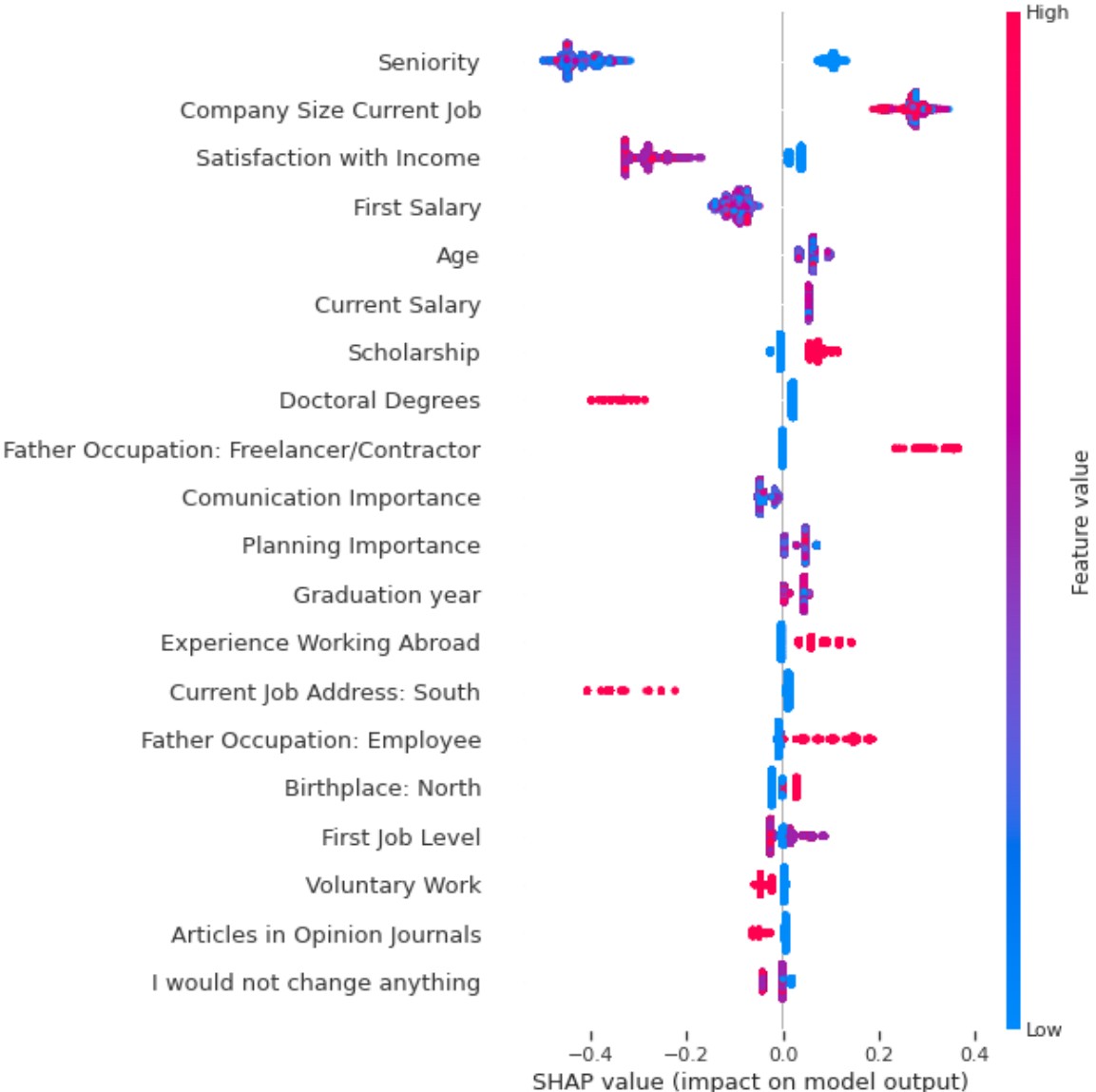

**Figure A1.** Most important features to predict a middle job level.

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
