# Peer review of "Analysis of Soft Skills and Job Level with Data Science: A Case for Graduates of a Private University"

_informatics, doi:10.3390/informatics10010023_

Round 1
Reviewer 1 Report
In this paper, the authors analyze which features are significant in predicting graduate employment levels and show the performance of three machine learning models in predicting graduate outcomes.
Decision trees, random forest and gradient boosting were used for data analysis, while ordinal regression was used as a benchmarking model.
The authors use unpublished data to analyze graduate outcomes from a survey of a private educational institution.
Gradient boosting turns out to be the best predictive model. In addition, current salary is the most important characteristic for predicting job level, followed by the importance of communication skills and teamwork, and then by the number of people in direct charge, company size, seniority, and satisfaction with actual income.
This work is an application of known models based on empirical data; in this regard, the authors should consider whether to share the data to ensure reproducibility.
The final judgment is: MINOR REVISIONS. More detailed comments will follow.
1. All prediction measures of the models considered are derived from the confusion matrix. The authors should consider using an additional score, such as Area Under Curve (AUC) or logloss, to compare the performance of the models.
2. The overall level of English is modest and there are too many typos. Therefore, in my opinion, the manuscript needs to be improved before resubmission; I strongly suggest a round of professional proofreading.
3. According to the reference style, only the number of references is necessary in the text, please be consistent throughout the document. For example, the citations on lines 26 and 27 do not follow the editorial style because the authors' names have been added before the numbering. The References section also needs to be revised.
4. The main concern is the categorization of some features. In particular, the occurrences of "job level" have been merged into 3 macro-levels. These new categories seem to be rather arbitrary; there is no clear explanation for the choice of putting some job positions in the same level. Above all, there seems to be no connection between the definition of occupational levels described in the introduction and the categorization adopted by the authors in the study. The authors need to better explain this part.
5. lines 204-211. The empirical description should be supported by an appropriate test, such as the chi-square, to see if there is a significant association between the two considered variables.
6. Table 3 should be deleted and the information on parameter settings of the considered models should be incorporated into the text.
7. The formatting style of Table 4 is not acceptable, please revise it and correct the typos in the table footnote.
8. The width of all the figures seems to be less than the width of the text, making it difficult to read the labels. Please resize them accordingly.
Author Response
Thanks for the comments and suggestions.
We add a file answering every comment. All the recommendations in missing explanations and new analysis were covered.

Reviewer 2 Report
The paper is devoted to ranking of attributes and their correlation, in the framewrok of dataset concering job position of graduates from private Mexican institution.
The main drawback of the paper is luck of detailed description of the methodology for the preparation of the dataset and performed experiments. They have not been sufficiently and clearly described.
Authors should explain why they chose such data mining algorithms?
In addition, they write on comparing classification results based on decision trees and regression.
So what type of data was used for the experiments? Since regression was used, was there real data? - if yes - what method of discretization was used for decision trees? What tree-building algorithm was used?
Authors write about 5-fold cross validation method and division of dataset into 70%- for train and 30%- for test – these information are inconsistent. Methodology of 5-fold cross-validtion is based in division of datset into 5 parts, and 1/5 is considered as a test part.
How the SHAP was used for selection of features – it is not described in the paper and how the classification results influence on the ranking of attributes?
Some information related to feature selection and creation of ranking of variables would be valuable.
Author Response
Thanks for the comments.
We add a file answering every comment. All the recommendations in missing explanations were covered in the manuscript.

Round 2
Reviewer 2 Report
After improvements paper can be accepted.